# Reactive Printing and Wash Fastness of Inherent Flame Retardant Fabrics for Dual Use

**DOI:** 10.3390/ma15144791

**Published:** 2022-07-08

**Authors:** Martinia Glogar, Tanja Pušić, Veronika Lovreškov, Tea Kaurin

**Affiliations:** Department of Textile Chemistry and Ecology, Faculty of Textile Technology, University of Zagreb, 10000 Zagreb, Croatia; tanja.pusic@ttf.unizg.hr (T.P.); veronika.lovreskov@ttf.unizg.hr (V.L.); tea.kaurin@ttf.unizg.hr (T.K.)

**Keywords:** FR fabric, printing, wash fastness, spectrophotometry

## Abstract

The possibility of reactive printability on protective flame—resistant fabrics, varied in composition of weft threads and weave was investigated. In addition, the wash fastness of printed samples was analyzed. The functional properties of fabrics were assessed by measuring of the Limiting Oxygen Index (LOI). Printing was performed with two printing pastes varied in thickeners and two dyestuff concentrations. The samples were analyzed by microscopic imaging using digital microscope and spectrophotometric measurement before and after the five washing cycles. The results confirmed the printability of FR inherent fabrics specified through fine colored effects and optimal wash fastness.

## 1. Introduction

Non-flammability of textiles has been the subject of intensive research since the 1940s, although there are also studies published as early as 1735 and 1821 on finishing of cellulosic textiles using alum, ferrous sulphate and borax, which laid the foundation for a modern approach to the research and development of non-flammable textiles [1,2,3]. There are two main directions in current research in the field of non-flammable textiles—research of flame-retardants of innovative structure and pre-treatment methodologies, and research into inherently non-flammable fibers, yarns and textiles. The most common flame retardants formulation developed in the second part of the 20th century was based on phosphorous, nitrogen or halogen derivate. For cotton, formulations based on organophosphorus compounds, such as tetrakis (hydroxymethyl) phosphonium chloride (THPC), hydroxyl functional organophosphorus oligomer (HFPO), dimethyloldihydroxylethyleneurea (DMDHEU) as well as some formulations based on monoguanidine dihydrogen phosphate (MGHP) and 3-aminopropylthoxysilane (APS), in some combinations with phosphoric acid were used [4,5,6,7,8,9,10,11,12,13]. The characteristics of such finishes applied on cellulose-based textiles can be changes in the comfort and physical-mechanical properties, as well as their negative environmental profile (e.g., release of free formaldehyde) [14,15,16]. The dynamics of changes in these properties depends on the characteristic of textile materials, process conditions, the composition of the bath, the method of application of functional agents (e.g., plasma, impregnation, LbL-layer by layer etc.) drying temperature and condensation [14]. In order to overcome such shortcomings, the direction of research is moving towards environmentally more acceptable and economically justified formulations with reduced amount of formaldehyde being released from the treated fabric, using butane tetra carboxylic acid (BTCA) as the binding agent [4]. In addition, some halogen-free phosphorus–nitrogen-based flame retardants were developed to promote more char formation during burning of the cellulosic substrate [17,18,19].

In 2011, Horrocks published a comprehensive study of innovative FR functionalization based on the application of nano-ZnO or TiO_2_, various clay compounds, and polycarboxylic acids for application to cotton [20,21]. In studies of low-pressure plasma pre-treatment in developing nano-coatings, Tsafack et al. reported an FR efficiency in grafting of phosphorus-containing acrylate monomers to polyacrylonitrile fabrics [22,23]. Jama et al. studied the possibility of using low pressure plasma in deposition of silicon-based nano-layer for improving the flame retardancy of polyamide 6 [24,25].

In the scope of inherent FR yarns and fabrics, intended for protective clothing, mixtures of fibers (synthetic and natural origin) are primarily applied, in order to achieve optimal non-combustibility, but also meet certain comfort standards. Sonee et al. tested the burning behavior of FR viscose and meta-aramid blended in three different ratios, as well as FR viscose with polyamide 6.6 and meta-aramid blended in two different ratios, confirming the best behavior of viscose/meta-aramid blends in ratio 30:70 [26,27]. One of the problems with blends with the purpose of achieving inherently non-combustible fabric is the nonlinearity of the final properties of the burning behavior to the structure of the blend [28]. Due to the high hydrophobicity and crystallinity of individual components of such blends, dyeing and printing is complex and requires a systematic approach to the selection of dyestuff, but also optimizing a key process parameter in dyeing or printing. Manyukov et al. have published an interesting study about dyeing of thermostable para/meta aramid by pre-treatment in a solvent-water-swelling system, and subsequent dyeing by the depletion process with a mixture of disperse and cationic dye [29].

Regarding printing on inherently flame-retardant fabrics, a review of the literature revealed a research gap and a relatively small number of publications. This was exactly the challenge and incentive in the design of the experiment, to examine, among other things, the characteristics of coloration that can be achieved given the limited content of components that can be dyed. The research is part of the project of the development of a functional fabric that has the properties of inherent flame retardancy and high comfort, whose visual and functional properties can be improved through printing.

## 2. Materials and Methods

Research was performed on five non-commercial fabrics whose composition and constructional properties have been designed and developed in cooperation of Croatian textile factory Cateks and University of Zagreb faculty of Textile Technology. Two fabrics were in twill 2/2, two in twill 3/1 and one in ripstop. The warp thread of these fabrics are the same, blend of meta-aramid, m-AR (95%) and para-aramid, p-AR (5%), while the weft threads are different in composition, polyamide (PA), FR viscose (CV FR) am meta-aramid (m-AR), Table 1.

Detailed characterization of structural and mechanical properties of fabrics is shown in Table 2.

Samples were woven on sample weaving machine Fanyuan Instrument DW598 (Hefei Fanyuan Instrument Co. Ltd., Hefei, Anhui, China), fully automated loom for weaving patterns with a woven rod. The loom is characterized with the maximum width of the base of 50 cm; number of wefts per minute of 30–60 and with maximum number of sheets of 20.

### 2.1. LOI (Limiting Oxygen Index)

The flame resistance of the new fabrics was tested by LOI—Limiting Oxygen Index. measured using Limiting Oxygen Index Apparatus (Concept Equipment Ltd., Arundel, UK) according to EN ISO 4589-2: 2017 Plastics—Determination of burning behavior by oxygen index—Part 2: Ambient—temperature test. Samples in size 140 mm × 53 mm were conditioned (RH 65%, 24 h) according to standard ISO 139:2008/A1 Textiles—Standard Atmospheres for Conditioning and Testing.

### 2.2. Micro Cone Calorimeter (MCC)

Microscale combustion calorimeter (MCC) from Govmark, Farmingdale, NY, USA was applied for the thermal characterization of five FR fabrics according to ASTM D7309-2007. The sample in mass of few milligrams was heated to a specified temperature using a linear heating rate of 1 °C/s in a nitrogen stream, flow rate of 80 cm^3^/min. The thermal degradation products were mixed with a 20 cm^3^/min oxygen stream.

### 2.3. Fabric Surface pH Measuring

Contact pH measurement of functional fabrics was tested using a contact electrode InLab Surface Pro-ISM^®^ device SevenCompact ™ Duo S213, Mettler Toledo, Greifensee, Switzerland.

### 2.4. Screen Printing

Printing with reactive dyes was performed by hand screen procedure. After drying and fixing of the prints, wash fastness was tested. Reactive dye and two thickeners were used for the preparation of printing pastes, each in two different viscosities defined by the ratio of water and dry matter. The first thickener was CHT-Alginat MV (CHT Group and Co) and the second Alkagum NS (Diamalt AG München 13, München, Germany). Both were prepared in 4% and 9% of a thickening agent (dry matter), so different viscosities were tested. The reactive dyestuff anthraquinone structure used was Brilliantblau V-R spez, Bezema (C.I. Reactive Blue 19, C.I. 61,200), in two concentrations. Pastes with a lower concentration of dye are marked with “a”, and pastes with a higher concentration of dye with “b”. The composition of the printing pastes is shown in Table 3. Quantities of components in printing pastes are shown per 100 g of printing paste. The samples were printed by hand screen procedure, dried and fixed with steam for 10 min.

### 2.5. Wash Fastness Testing and Spectrophotometric Measurement

Printed samples were tested in a laboratory apparatus Polycolor, Mathis, Oberhasli, Swiss. The test was performed according to standard ISO 105-C06:2010 (A2S) Textiles—Tests for color fastness—Part C06: Colour fastness to domestic and commercial laundering, using 5 g/L standard detergent (ECE Non phosphate detergent without optical brightener agent), with bath ratio 1:8, temperature 40 ± 2 °C, time 30 min, through 5 cycles. The samples were air dried between each cycle.

The wash fastness of prints was assessed by spectral evaluation of samples before and after washing, using remission spectrophotometer DataColor 850 (Datacolor AG, Lucerne, Switzerland), with constant instrument aperture, standard light D65 and d/8°geometry. The results are shown in terms of color difference values calculated according to CIE76 formula.

### 2.6. Microscopic Imaging

The microscopic examination of a printed samples was performed using DinoLite AM7013 under magnification of 50×. The imaging was performed before and after the 5th cycle of washing and drying. The microscopic imaging has been performed with the following parameters: magnification: 50×/1.3 MP; unit: mm; horizontal FOV [accuracy]: 9.564 mm [+/−0.192 mm]; one pixel increment (one keyboard arrow press): ~7.4 μm).

The scanning of the fabrics was performed with resolution of 1200 dpi.

## 3. Results

FR properties of fabrics before printing in warp and weft direction by LOI-Limiting Oxygen Index were examined according to a standard protocol are presented in Table 4.

The results of LOI in Table 4 indicate on high resistance to burning, so all samples meet the criteria specified for the inherently flame-resistant fabrics with LOI > 26%. Comparison of LOI values did not show significant differences within variations in weft and weave. 

MCC as a method for evaluation of the fire retardancy of materials using only a few milligrams of sample showed small differences between samples, Table 5.

Tested fabrics can be grouped according to values of MCC parameters as follows. The heat release capacity (ηc) of fabrics ranges from 39.00 [J(g·K)^−1^] for fabric 2 to 51.00 [J(g·K)^−1^] for fabric 5. Other fabrics are specified between these values. Fabric 1, Fabric 2 and Fabric 4 possessed more optimal values than Fabric 3 and 5. Peak heat release rates (Qmax) for selected fabrics 1, 2 and 4 fully follow the values of the heat release capacity. Specific heat release of samples (hc) fabric 1 and fabric 3 is less than 7.00 [kJ·g^−1^], while to fabric 2 and fabric 5 possessed the same value, 7.20 [kJ·g^−1^], and finally the fabric 5 to which belongs the highest value of 8.00 [kJ·g^−1^]. Pyrolysis residues of all tested fabrics are higher than 40%. Finally, according to MCC criteria small differences were found between tested fabrics, which can be attributed to structural features, primarily in the fabric embroidery. Nevertheless, all fabrics can be specified as fire retardant.

The surface of fabrics was characterized by pH value, which may indicate on character of residual substances, Table 6.

The values obtained indicate acidity of the surface which may be caused by yarn preparations. The acidic surrounding can affect the reduced bonding of the reactive dye with the fiber, since it requires alkaline conditions to induce the reaction of the dye with the fiber. So samples were before printing washed in a mild detergent composed from anionic and non-ionic surfactants. After the analyses of FR properties of the fabrics along with the characterization of the surface pH, the process of printing with reactive dye was approached.

Figure 1, Figure 2, Figure 3, Figure 4 and Figure 5 shows microscopic images of printed fabrics. It was observed that the binding of dyes in the process of printing and fixing occurs exclusively on the viscose components of the yarn. The difference in the depth of blue coloration, that is visually noticeable in samples Fabric 1, Fabric 2 and Fabric 4, compared to samples Fabric 3 and Fabric 5, stems from the difference in the basic color of fabrics, which has a grey component in weft yarn. The images of the samples serve the visual orientation of the appearance of the achieved coloration. In addition, the quality of the print is, in general, assessed by the sharpness of the print and the achieved color characteristics.

For Fabric 1, for samples printed with Paste 2a, 4a and 4b, satisfactory print sharpness was obtained (Figure 1).

The result indicates that a thickener with a higher dry matter content is suitable for this type of fabric with a hydrophobic fiber content (aramids and PA), as in the case of the indicated pastes, regardless of the type of thickener. The microscopic images clearly show the component of aramid yarn that remained uncolored, and the proportion of the same is visible due to the characteristic of the weaving. Since the appearance of the color depends on the interaction of simultaneous remission from the dyed and undyed components of the fabric, an influence of the uncolored component on obtained color appearance can be expected.

Fabric 2 (Figure 2) differs from Fabric 1 in the type of weaving, which also affects the behavior of the printing paste.

It can be noticed that for Paste *a*, regardless of the type of thickener and the amount of dry matter, satisfactory print sharpness was obtained, but for Paste *b* where a higher amount of dye is present (recipes in Table 3), some printing paste spreading outside the pattern cotour occurred for thickener CHT-Alginate MV. A starch-based thickener (Alkagum NS) proved to be more suitable for printing Fabric 2 with higher dye concentration. 

Fabric 3 (Figure 3) is of the same weave as Fabric 2, differing in the proportion of PA fibers (Table 1). Furthermore, in contrast with Fabrics 1 and 2, Fabric 3 has completely uncolored components in its composition and does not already contain the basic gray shade. That is why a clearer blue color was obtained.

A similar trend of differences is observed as for Fabric 2. For Paste *b*, which is characterized by a higher proportion of dyestuff, spreading of dye outside the print contour is observed. On microscopic images it is observed for samples printed with the same Paste *b*, a darker color of the dye-binding component, which is associated with the occurrence of dye capillary spreading. For the component that has an affinity for the reactive dye, which is viscose, there was saturation, i.e., maximum dye binding, and the excess dye caused capillary spillage. 

For fabric 4 (Figure 4), printed with Paste *a*, with a lower dye content, satisfactory print sharpness was obtained for recipes with a higher viscosity thickener meaning a higher dry matter content (Paste 2a and 4a), regardless of the type of thickener.

As for Pastes *b* with a higher dye content, again a satisfactory print sharpness is obtained for printing pastes 3 and 4 with a starch-based thickener (alkagum NS). Microscopic images clearly show the difference in weaving structure, which results in a difference in the ratio of dyed and undyed components in the fabric content, and the impact of such yarn ratio on the subjective color appearance as well as on the objective measurements that follow, is expected.

For Fabric 5 (Figure 5), for Pastes *a*, with a lower content of dye, satisfactory, given the sharpness of the print contour, proved to be pastes with a thickener of higher viscosity, ie a higher proportion of dry matter (2a, 4a), regardless of the type of thickener. When printing with pastes with a higher proportion of dye, capillary spillage occurs, with the exception of the sample printed with Paste 4b (paste with a starch-based thickener, higher viscosity).

Paste 1a showed the stability in viscosity and homogeneity of the structure and sharp, equal prints were achieved without spillage and capillary migration. The same effects were achieved for pastes 2a, 3b and 4b. Pastes 1b, 3a and 4a did not gave satisfactory print sharpness, although the coverage of surface with printing paste is of optimal uniformity. In addition, during printing and fixing, capillary spreading of the paste occurred, although there were no changes in the viscosity and homogeneity of the paste during the process. Paste 2b also did not achieve satisfactory print sharpness, although the coverage of the surface with printing paste was optimal here as well.

Comparative analysis is given of the objective values of color strength (K/S) and the ratio of lightness (L*) and chroma (C*) as a definition of color intensity. The results of K/S are shown graphically in Figure 6a–e, and the L*/C* values are shown in Table 7.

Objective K/S values show a relatively low achieved color strength. As expected,. slightly higher values were obtained for printing pastes “b” with a higher concentration of dyestuff. Such lower K/S values were to be expected, given that only a partial ratio of components contained in the yarn composition is capable of bonding with dyestuff (the cellulosic part contained in the viscose yarn component).

The Kubelka-Munk coefficient, which objectively evaluates the color strength (K/S), is defined by the amount of dye that binds to the fiber in the printing process. The higher the color strength coefficient K/S, the greater the color depth itself, which estimates the coverage of the surface with color in the printing process. The printing process in which dyes are applied is actually a dyeing process from small, concentrated baths that takes place in the fixing phase. In this process, a certain amount of dye binds to the fiber, which depends on the conditions created by the thickener and other components in the printing paste during the fixing phase.

However, the K/S color intensity data alone does not provide information on the appearance of the achieved color. The K/S value needs to be analyzed in the context of the ratio of lightness (L*) and chroma (C*) for a given tone (h°). Namely, the same values of color strength will not have the same interpretation for different color hues and will depend on the understanding of the nature of color. According to its nature, blue belongs to naturally darker colors, which means that maximum chroma is achieved at lower lightness, expressed in objective CIE values, at lightness levels below 50. Therefore, the results indicated a lower color intensity, given the obtained lightness values above 50 (except for fabric 4) and relatively low saturation values. Such ratio of lightness (L*) and chroma (C*) is in accordance with the K/S values. The specific ratio of lightness and chroma, in general, is the definition of color intensity and depends, in addition to the parameters of the dyestuff and the textile material, also on the color itself.

In print characterization, color reproduction quality control is performed by evaluating the spectrophotometric parameters. In dye printing, achieving a satisfactory intensity and brilliance of coloration of printed parts of the fabric is very complex. Precisely due to the chemical bonding of the dye with the fiber, the obtained coloration becomes an integral part of the fabric structure, which provides more satisfactory properties of fastness, but lower brilliance of the dye. It is the color intensity as well as the brilliance of the print coloration that can be defined by the specific ratio of lightness (L*) and chroma (C*). In the samples tested in this paper, the lower intensity of the obtained coloration is contributed by the specific composition of the fabric in which more than half of the content are fibers that cannot be dyed by the process of dye-fiber chemical bonding.

The spectrophotometrically measured values of samples after the 5th wash cycle were compared with the values of unwashed samples and the analysis of the change in the value of K/S (color strength) and the specific relationship of lightness (L*) and chroma (C*) was performed. The results for K/S are shown graphically on Figure 7 and the results for lightness (L*) and chroma (C*) are given in Table 8.

For K/S values, a minimal change was obtained, after the washing cycles, which indicate that there were no evident release of dye. Slightly higher differences in K/S values after washing were obtained for Fabric 4 printed with Paste b, regardless of the type and viscosity of the thickener. It can be said that, in this sample, the amount of bound dye was lower and there was a higher release of dye from during washing.

For the ratio of lightness (L*) and chroma (C*), in general for all samples (Fabrics 1 to 5), it was observed that there was an increase in lightness (L*) and a decrease in chroma (C*). This result of the lightness-chroma (L*-C*) ratio is in line with the K/S values and the indication of a slight dye release. An increase in lightness and a decrease in chroma indicates a shift in coloration towards a lighter, more pastel color shade, indicating albeit slight but still fading coloration in the wash.

In further analysis, the color differences are calculated based on objective spectrophotometric measurement of samples before and after the 5 washing cycles, taking the values of printed unwashed samples as the standard (reference) values. The results are shown in terms of color parameters differences (lightness difference dL*, chroma difference dC* and hue difference dh) as well as in terms of total color difference value (dE), presented graphically in Figure 8.

For the parameters of lightness (L*) and hue (h°), the differences (dL* and dh) were for all samples within the ranges of tolerances (tolerances for dC* = 0.8–1.5; dL* = 1.2–2; dh = 0.5–0.8). The differences obtained for chroma parameter (dC*) were higher and outside the tolerances. However, although the individual differences in chroma (dC*) were out of the tolerances, the values of the total color difference (dE), taking into account the obtained minimum differences in lightness and hue, were also mostly within the tolerance limits. More pronounced values of the total color difference (dE), outside the range of tolerance (dE < 1.5), were obtained only for Fabric 1, Paste 3b. The results of the differences are acceptable and indicate a satisfactory color wash fastness.

## 4. Conclusions

Based on the analysis, it can be confirmed that it is possible to achieve a certain level of coloration in fabrics that contain a high proportion of aramid fibers (95% in warp and 40% in weft yarn composition), if a certain ratio of cellulose component is contained (30–40% in weft yarn). The weft yarn contained a component of viscose (38–40%) that has the ability to bond reactive dye, and coloration was achieved even in samples that had a certain basic color i.e., were not completely uncolored (Fabrics 1, 2 and 4). Although, by objective evaluation of the color strength, lower K/S values were obtained (1.1 to 2.6), by analyzing the relationship between the color lightness parameters (L*) and chroma (C*), a satisfactory color intensity was achieved (maximum values L* = 62.31 and C* = 21.78).

The results confirmed the optimal wash fastness in the process of five washing cycles, more emphasized for Fabric 5, where for all printing conditions (sort and viscosity of the thickener as well as dyestuff content) the lowest differences values have been obtained (dE_CIE_ = 0.32–0.91).

This research is part of the comprehensive research of the characteristics of FR fabrics and the possibility of their finishing in the processes of dyeing and printing.

## Figures and Tables

**Figure 1 materials-15-04791-f001:**
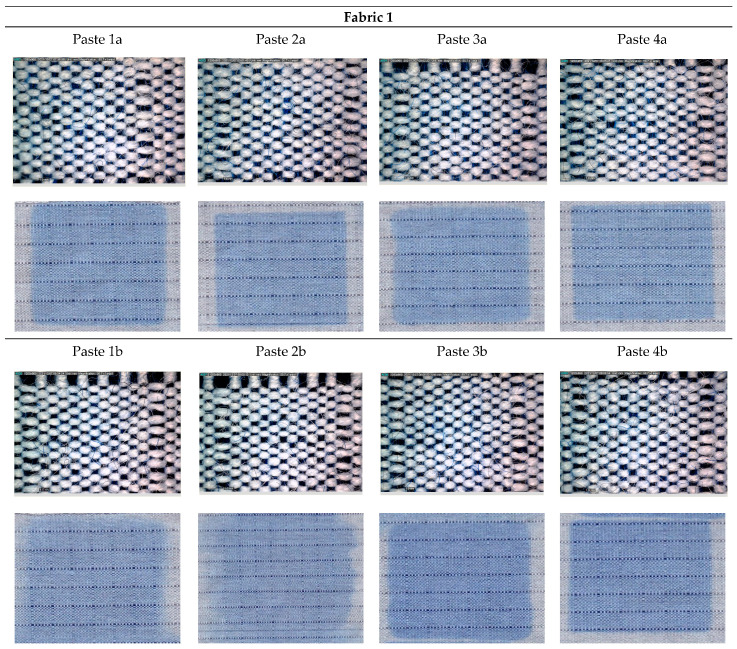
DinoLite microscopic and scanned images of printed Fabric 1.

**Figure 2 materials-15-04791-f002:**
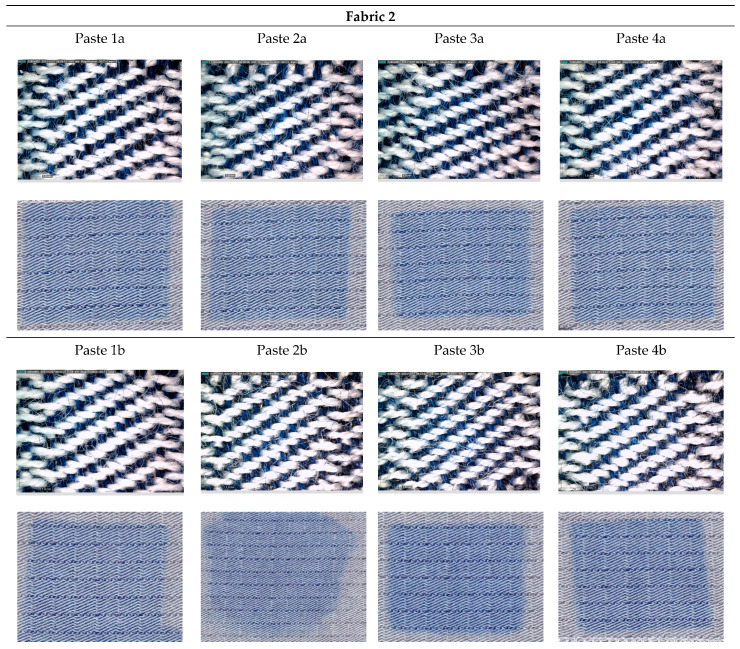
DinoLite microscopic and scanned images of printed Fabric 2.

**Figure 3 materials-15-04791-f003:**
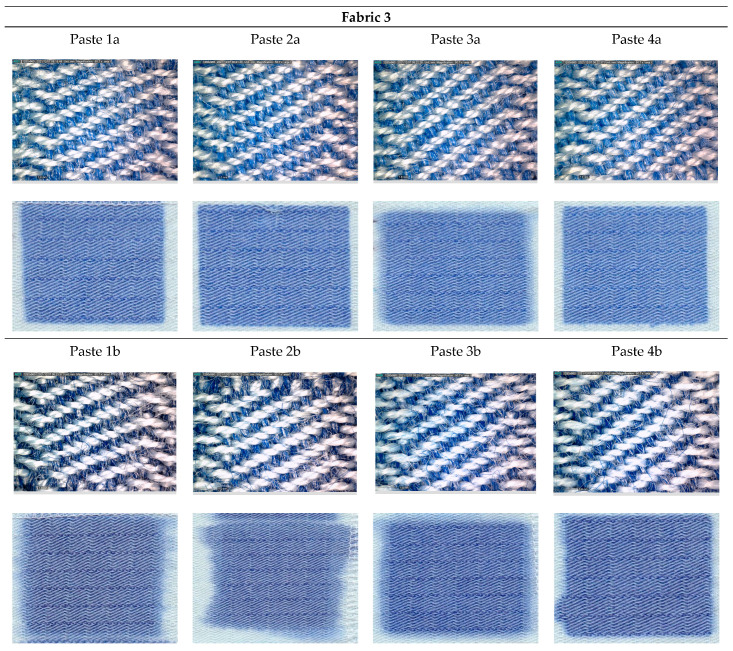
DinoLite microscopic and scanned images of printed Fabric 3.

**Figure 4 materials-15-04791-f004:**
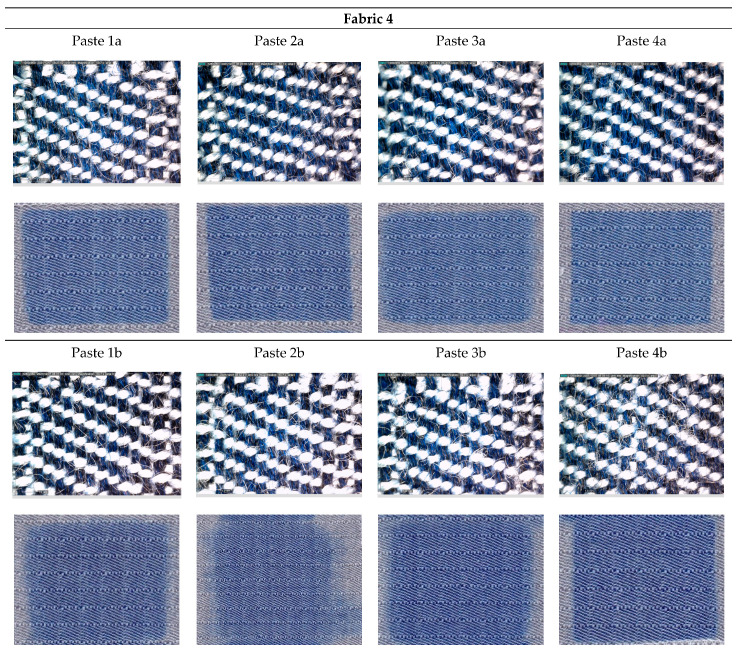
DinoLite microscopic and scanned images of printed Fabric 4.

**Figure 5 materials-15-04791-f005:**
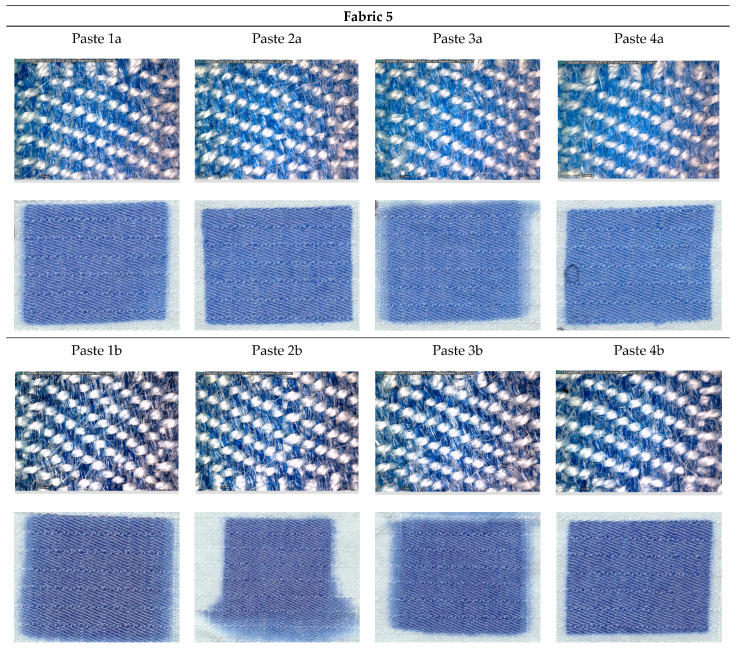
DinoLite microscopic and scanned images of printed Fabric 5.

**Figure 6 materials-15-04791-f006:**
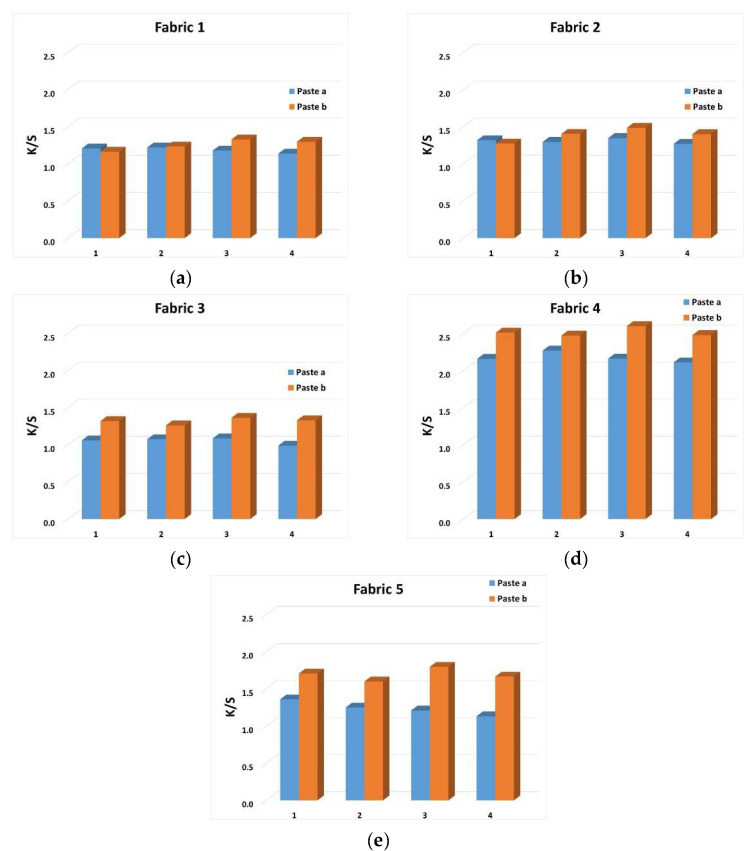
K/S objective values of color of printed surfaces: Fabric 1 (**a**), Fabric 2 (**b**), Fabric 3 (**c**), Fabric 4 (**d**) and Fabric 5 (**e**).

**Figure 7 materials-15-04791-f007:**
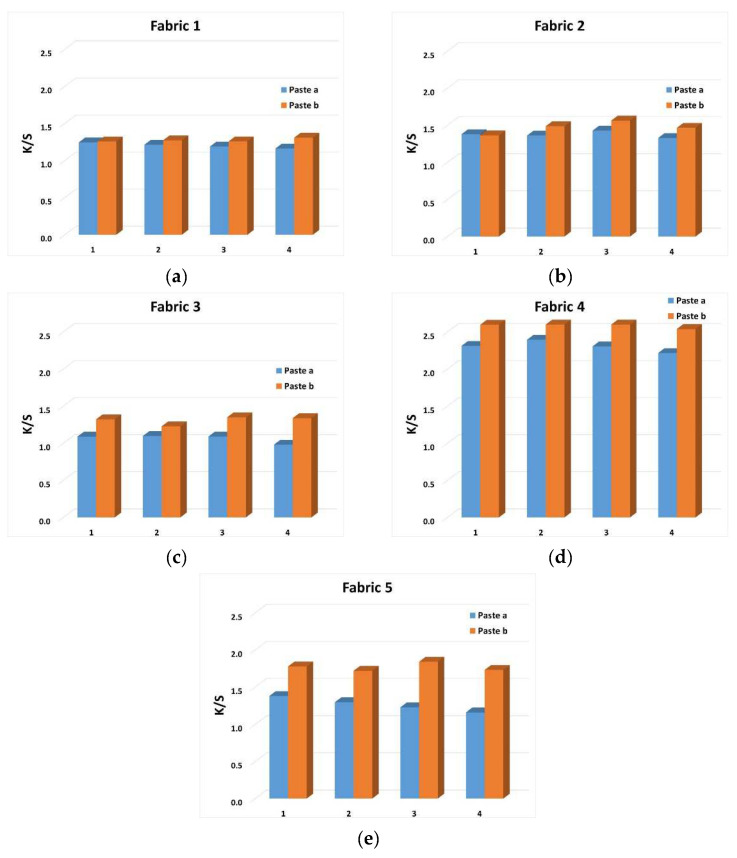
K/S objective values of color of printed surfaces after 5 washing cycles: Fabric 1 (**a**), Fabric 2 (**b**), Fabric 3 (**c**), Fabric 4 (**d**) and Fabric 5 (**e**).

**Figure 8 materials-15-04791-f008:**
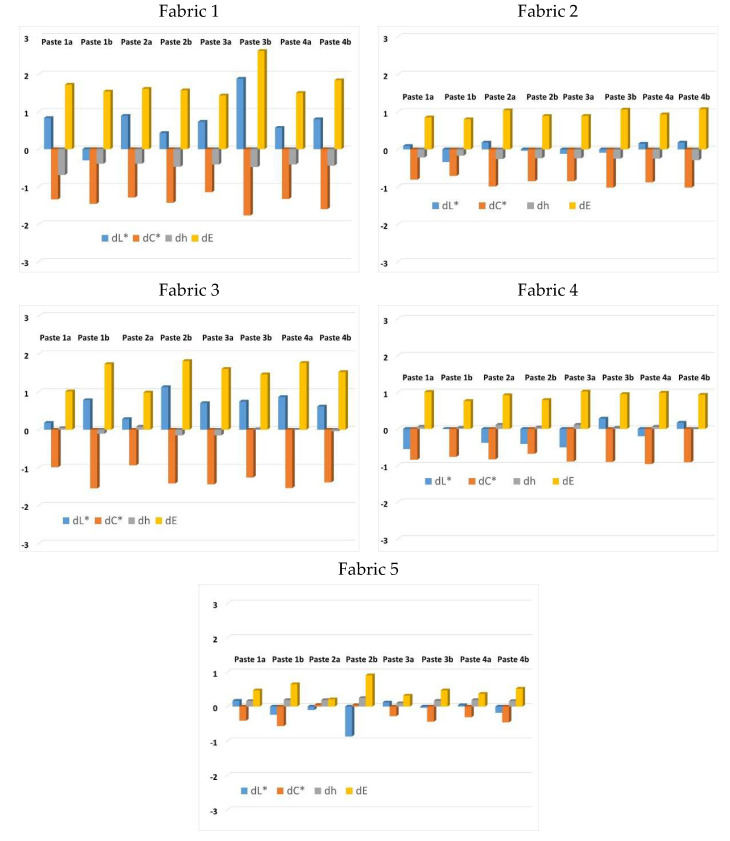
Color differences values calculated according to CIE76 formula. comparing values of unwashed samples with values obtained after 5th wash cycle.

**Table 1 materials-15-04791-t001:** Composition of fabrics.

	Fabric 1	Fabric 2	Fabric 3	Fabric 4	Fabric 5
Weave	Ripstop	Twill 2/2	Twill 2/2	Twill 3/1	Twill 3/1
Warp	95% m-AR	95% m-AR	95% m-AR	95% m-AR	95% m-AR
	5% p-AR	5% p-AR	5% p-AR	5% p-AR	5% p-AR
Weft	2% PA	2% PA		2% PA	
	20% PA 6.6	20% PA 6.6	20% PA6.6	20% PA 6.6	20% PA6.6
	38% CV FR	38% CV FR	40% CV FR	38% CV FR	40% CV FR
	40% m-AR	40% m-AR	40% m-AR	40% m-AR	40% m-AR

**Table 2 materials-15-04791-t002:** Constructional and mechanical characteristics of fabrics.

	Mass [g/m^2^]	Thickness 0.5 kPa [mm]	Thickness 1 kPa [mm]	Warp Density [Threads/cm]	Weft Density [Threads/cm]
Fabric 1	203	0.66	0.63	38	20
Fabric 2	197	0.78	0.75	37	20
Fabric 3	223	0.80	0.78	37	20
Fabric 4	195	0.84	0.82	36	20
Fabric 5	215	0.91	0.88	37	20

**Table 3 materials-15-04791-t003:** Composition of printing pastes.

Paste	Thickener		Dyestuff	Urea	Na_2_CO_3_
1a	CHT-Alginat MV (4%)	50 g	1.26 g	20 g	4 g
2a	CHT-Alginat MV (9%)	50 g	1.26 g	20 g	4 g
3a	Alkagum NS (4%)	50 g	1.26 g	20 g	4 g
4a	Alkagum NS (9%)	50 g	1.26 g	20 g	4 g
1b	CHT-Alginat MV (4%)	50 g	7.5 g	20 g	4 g
2b	CHT-Alginat MV (9%)	50 g	7.5 g	20 g	4 g
3b	Alkagum NS (4%)	50 g	7.5 g	20 g	4 g
4b	Alkagum NS (9%)	50 g	7.5 g	20 g	4 g

**Table 4 materials-15-04791-t004:** LOI of fabrics.

Fabric Sample		LOI [%]		
	Warp	σ [%]	Weft	(σ) [%]
Fabric 1	33.6	0.266	33.55	0.266
Fabric 2	34.2	0.191	33.96	0.266
Fabric 3	34.4	0.191	34.37	0.151
Fabric 4	34.1	0.110	34.10	0.261
Fabric 5	34.68	0.151	34.68	0.110

**Table 5 materials-15-04791-t005:** MCC parameters.

Parameters	Fabric 1	Fabric 2	Fabric 3	Fabric 4	Fabric 5
ηc (J[g·K]^−1^)	43.00	39.00	50.00	44.0	51.00
Qmax [W·g^−1^]	42.68	38.71	50.73	41.56	51.72
hc [kJ·g^−1^]	6.60	7.20	6.40	8.00	7.20
hc, gas [kJ·g^−1^]	12.30	12.90	11.46	13.74	12.19
Tmax [°C]	281.9	276.0	281.1	279.0	279.3
Residue [%]	46.33	44.17	44.16	41.79	40.93

**Table 6 materials-15-04791-t006:** pH evaluated by contact electrode.

Sample	pH	(σ)	T [°C]
Fabric 1	4.57	0.246	25.5
Fabric 2	4.73	0.380	25.7
Fabric 3	4.59	0.249	25.7
Fabric 4	4.51	0.264	24.9
Fabric 5	4.40	0.452	24.5

**Table 7 materials-15-04791-t007:** Lightness (L*) and chroma (C*) relationship for printed samples. as definition of color intensity.

	**L***	**C***		**L***	**C***
Fabric 1 Paste 1a	59.83	10.11	Fabric 1 Paste 1b	59.79	9.83
Fabric 1 Paste 2a	59.73	9.87	Fabric 1 Paste 2b	59.29	9.18
Fabric 1 Paste 3a	59.94	10.36	Fabric 1 Paste 3b	56.80	12.48
Fabric 1 Paste 4a	60.71	9.92	Fabric 1 Paste 4b	57.86	11.56
	**L***	**C***		**L***	**C***
Fabric 2 Paste 1a	58.61	8.47	Fabric 2 Paste 1b	59.68	7.34
Fabric 2 Paste 2a	58.57	8.68	Fabric 2 Paste 2b	57.75	7.86
Fabric 2 Paste 3a	57.82	9.48	Fabric 2 Paste 3b	55.65	10.94
Fabric 2 Paste 4a	58.94	8.53	Fabric 2 Paste 4b	57.32	9.09
	**L***	**C***		**L***	**C***
Fabric 3 Paste 1a	60.22	17.76	Fabric 3 Paste 1b	56.84	13.28
Fabric 3 Paste 2a	60.56	18.34	Fabric 3 Paste 2b	57.97	13.86
Fabric 3 Paste 3a	59.90	18.94	Fabric 3 Paste 3b	55.48	17.70
Fabric 3 Paste 4a	61.45	18.95	Fabric 3 Paste 4b	56.42	16.53
	**L***	**C***		**L***	**C***
Fabric 4 Paste 1a	48.58	11.85	Fabric 4 Paste 1b	46.28	10.41
Fabric 4 Paste 2a	47.90	11.68	Fabric 4 Paste 2b	47.05	10.07
Fabric 4 Paste 3a	48.40	12.06	Fabric 4 Paste 3b	44.93	13.03
Fabric 4 Paste 4a	48.66	12.30	Fabric 4 Paste 4b	46.20	11.85
	**L***	**C***		**L***	**C***
Fabric 5 Paste 1a	55.27	20.02	Fabric 5 Paste 1b	51.51	14.24
Fabric 5 Paste 2a	56.20	21.19	Fabric 5 Paste 2b	53.02	15.13
Fabric 5 Paste 3a	56.60	21.10	Fabric 5 Paste 3b	50.41	17.96
Fabric 5 Paste 4a	57.48	22.08	Fabric 5 Paste 4b	51.53	18.26

**Table 8 materials-15-04791-t008:** Lightness (L*) and chroma (C*) relationship for printed samples after 5 washing cycles.

	**L***	**C***		**L***	**C***
Fabric 1 Paste 1a	60.66	8.77	Fabric 1 Paste 1b	59.48	8.36
Fabric 1 Paste 2a	60.61	8.58	Fabric 1 Paste 2b	59.72	7.74
Fabric 1 Paste 3a	60.67	9.21	Fabric 1 Paste 3b	58.68	10.71
Fabric 1 Paste 4a	61.28	8.59	Fabric 1 Paste 4b	58.66	9.96
	**L***	**C***		**L***	**C***
Fabric 2 Paste 1a	58.69	7.66	Fabric 2 Paste 1b	59.34	6.64
Fabric 2 Paste 2a	58.75	7.69	Fabric 2 Paste 2b	57.71	7.01
Fabric 2 Paste 3a	57.70	8.63	Fabric 2 Paste 3b	55.56	9.91
Fabric 2 Paste 4a	59.10	7.65	Fabric 2 Paste 4b	57.50	8.08
	**L***	**C***		**L***	**C***
Fabric 3 Paste 1a	60.40	16.76	Fabric 3 Paste 1b	57.62	11.73
Fabric 3 Paste 2a	60.84	17.40	Fabric 3 Paste 2b	59.08	12.45
Fabric 3 Paste 3a	60.59	17.50	Fabric 3 Paste 3b	56.22	16.44
Fabric 3 Paste 4a	62.31	17.41	Fabric 3 Paste 4b	57.03	15.14
	**L***	**C***		**L***	**C***
Fabric 4 Paste 1a	48.03	11.01	Fabric 4 Paste 1b	46.28	9.65
Fabric 4 Paste 2a	47.53	10.85	Fabric 4 Paste 2b	46.64	9.39
Fabric 4 Paste 3a	47.90	11.18	Fabric 4 Paste 3b	45.20	12.13
Fabric 4 Paste 4a	48.46	11.33	Fabric 4 Paste 4b	46.37	10.94
	**L***	**C***		**L***	**C***
Fabric 5 Paste 1a	55.44	19.61	Fabric 5 Paste 1b	51.26	13.67
Fabric 5 Paste 2a	55.92	21.25	Fabric 5 Paste 2b	52.15	15.17
Fabric 5 Paste 3a	56.72	20.82	Fabric 5 Paste 3b	50.37	17.51
Fabric 5 Paste 4a	57.52	21.78	Fabric 5 Paste 4b	51.36	17.80

## Data Availability

Data available in a publicly accessible repository.

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
