# Peer review of "Reactive Printing and Wash Fastness of Inherent Flame Retardant Fabrics for Dual Use"

_materials, 2022, doi:10.3390/ma15144791_

Round 1

Reviewer 1 Report

The presented work relies on the preparation of flame retarded polymer based fabrics able to be printed with reactive dyes. The novelty/originality of the work is not well underlined whereas the scientific design is insufficient. Moreover, an extensive editing of punctuation must be performed. In the following, some major revision that must be done before the publication:

-Introduction must focus of printable fabric and should include recent references. Moreover, it must underline the novelty and the originality of the work.

-since the substrate is not commercial, Authors should provide a deep characterization of the obtained fabric. Fabrication methodology should be also included in the text.

-Table 1 acronyms are not defined. The same is valid for Table 2

-Table 3 and 4 must report deviations, moreover double dots are not scientifically valid.

-Line 134-135 please explain why washing process should change the pH and why this parameter can affect the printability of the fabric

-Figures 1 and 3 must include scales of pictures.

-Line 156-159 Authors should explain this behaviour.

-Please explain the use of K/S, L and C parameters and which aspect printing characterize.

-Flammability test and cone calorimetry must be addressed in order to estimate the real flame retardant properties of samples.

Author Response

Dear Reviewer 1,
We thank you for the very useful suggestions that we have adopted and we believe that they have significantly contributed to the progress and quality of paper.
Detailed answer, point by point, is attached as a separate document.

Reviewer 2 Report

In the attached document are many suggestions/requirements.

Author Response

Dear Reviewer 2,
We thank you for the very useful suggestions that we have adopted and we believe that they have significantly contributed to the progress and quality of work.
Detailed answer, point by point, is attached as a separate document.

Round 2

Reviewer 1 Report

Dear Authors,

after the first revision the paper is improved but still needs some corrections before the publication, so I suggest a minor revision. In the following my recommendation:

lines 62-65 nano-coating instead on Nano-coating and nano-layer instead of Nano-layer

lines 86-90 must stay in Funding, this is not state of the art

lines 146-149 must stay on 2.1 section

table 5: explain and comment all the reported parameters for MCC. Moreover, FR properties could be tested by flammability test

figure 1 and similar: microscopic imagines characterization parameters must stay in 2.6 section

conclusion: lines 422-424. Authors could be more precise with quantities

Reviewer 2 Report

Thank you for making the revisions which have improved the paper.

Table 2 - round fabric mass; clarify what is meant by niti

Figures 1-4 - the scale information needs to appear on the image
